# Geometric Analysis of Three-Dimensional Woven Fabric with in-Plane Auxetic Behavior

**DOI:** 10.3390/polym15051326

**Published:** 2023-03-06

**Authors:** Muhammad Zeeshan, Hong Hu, Ehsan Etemadi

**Affiliations:** 1School of Fashion and Textiles, The Hong Kong Polytechnic University, Hong Kong, China; 2Research Institute for Intelligent Wearable Systems, The Hong Kong Polytechnic University, Hong Kong, China; 3Department of Mechanical Engineering, Hakim Sabzevari University, Sabzevar 9617976487, Iran

**Keywords:** auxetic woven fabrics, 3D textile structure, negative Poisson’s ratio, geometrical analysis

## Abstract

Auxetic textiles are emerging as an enticing option for many advanced applications due to their unique deformation behavior under tensile loading. This study reports the geometrical analysis of three-dimensional (3D) auxetic woven structures based on semi-empirical equations. The 3D woven fabric was developed with a special geometrical arrangement of warp (multi-filament polyester), binding (polyester-wrapped polyurethane), and weft yarns (polyester-wrapped polyurethane) to achieve an auxetic effect. The auxetic geometry, the unit cell resembling a re-entrant hexagon, was modeled at the micro-level in terms of the yarn’s parameters. The geometrical model was used to establish a relationship between the Poisson’s ratio (PR) and the tensile strain when it was stretched along the warp direction. For validation of the model, the experimental results of the developed woven fabrics were correlated with the calculated results from the geometrical analysis. It was found that the calculated results were in good agreement with the experimental results. After experimental validation, the model was used to calculate and discuss critical parameters that affect the auxetic behavior of the structure. Thus, geometrical analysis is believed to be helpful in predicting the auxetic behavior of 3D woven fabrics with different structural parameters.

## 1. Introduction

Auxetic textile is a sub-class of meta-materials that are differentiated from conventional materials due to their negative Poisson’s ratio (NPR) [1]. So far, scientists have successfully developed fibers [2,3], polymers [4,5], yarns [6,7,8], and fabrics [9,10] with the uncommon property of an NPR. Textile fabrics with an NPR mean that they expand when they are subjected to tensile loading and shrink when compressive loading is applied. Due to this unusual response to the applied loading, auxetic fabrics show outstanding extensibility in both warp and weft directions, better drapability, and become more compact upon compression [1]. Therefore, auxetic fabrics are considered to be the first choice for applications in protective textiles, medical textiles, smart textiles, sportswear, and auxetic composites [11]. Researchers have conducted an immense amount of work to develop auxetic fabrics in every possible way. Therefore, auxetic fabrics could be realized in woven form two-dimensional (2D) auxetic woven fabrics [12,13,14], three-dimensional (3D) auxetic textiles [15,16,17], knitted form (warp-knitted auxetic fabrics [18,19], weft-knitted auxetic fabrics [20,21]), non-woven form [22,23], and auxetic laminated fabrics [24,25]. Mainly, there are two techniques for introducing auxetic property into textile fabrics. The first and more simple technique is to use auxetic yarns and a conventional weave pattern. This technique was first introduced by Miller et al. [6], and they used their invented double helix yarn (DHY) in weft direction of a plain-woven fabric. The reason for using the DHY in weft direction was to ensure the out-of-register position of each neighboring DHY strand to maximize the auxetic effect. When the fabric was subjected to tensile tension, the DHY strands overlap each other in the thickness direction, which causes an out-of-plane auxetic effect by increasing the thickness of the sample. In this study, the maximum achieved out-of-plane Poisson’s ratio (PR) was −0.1. Similarly, the same research group also developed auxetic plain-woven fabrics using helical auxetic yarn (HAY) in warp direction and multi-filament polyester yarn in weft direction [26]. This study concluded that the auxetic effect is associated with the weft yarn’s material and weave geometry. Recently, auxetic woven fabrics were produced using auxetic plied yarns (4 ply and 6 ply) and DHY [27]. For fabric production, auxetic yarns were used in the warp direction only, while conventional elastic yarn was used in the weft direction. Three weave designs (plain, twill, and satin) were selected, along with other design parameters of plied yarn, to evaluate their effect on the NPR of the woven fabrics. Although producing auxetic woven fabrics with this method is the easiest way, there are still many drawbacks that limit the practical application of such fabrics. Those drawbacks are: the NPR of auxetic yarn transfers partially to the fabric due to weaving constraints, the unstable auxetic effect, and the out-of-register orientation of auxetic yarns. Furthermore, the orientation of auxetic yarn inside the fabric structure is crucial and difficult, i.e., the yarns should be as straight as possible to obtain a better auxetic effect. Due to this limitation, this technique is only suitable for woven fabrics and cannot be applied to knitted fabrics.

The second technique of developing auxetic fabrics includes the use of non-auxetic yarns and the realization of special auxetic geometry. It is well known that the auxetic effect is associated with the geometrical arrangement of a structure’s unit cells such as re-entrant hexagonal, rotating squares, etc. [28]. Therefore, the idea of developing auxetic fabrics progressed when scientists gave considerable attention to configuring conventional yarns in a special geometrical arrangement using knitting or weaving technologies. Up to now, various weft-knitted [20] and warp-knitted [18] auxetic fabrics have been developed using this technique. However, the low structural stability and low elastic recovery of the knitted auxetic fabrics make them impractical for many applications such as protective textiles, etc. Recently, a considerable amount of work has been reported on the development of 2D auxetic woven fabrics. For the first time, a uni-stretch auxetic woven fabric was developed using conventional weaving techniques and non-auxetic yarns [9]. Auxetic behavior was introduced to fabrics through foldable structures, re-entrant hexagonal geometry, and a rotating rectangle structure. The basic mechanism involved in the auxetic effect is the different shrinkage properties of tight weaves (plain) and loose weaves (twill or satin), along with the use of elastic and non-elastic yarns. These fabrics showed a zero PR or a very low NPR in one direction only. Therefore, another study was conducted by Zulifqar et al. [29] on developing bi-stretch auxetic woven fabrics. The methodology of the study was based on a similar principle of differential shrinkage, but elastic yarns were used in both the warp and weft directions along with non-elastic yarns. Among all the developed auxetic structures based on different geometries, the zig-zag folded stripes showed the highest NPR of −0.15. The developed 2D auxetic woven fabrics have the potential to be used for clothing, fashion garments, etc. However, these fabrics cannot be used for high-performance applications because they experience more longitudinal deformation under tension, a low NPR, and a sharp decrease in the NPR under increased strain.

Nowadays, 3D auxetic fabrics have been the central focus for many researchers because they can exhibit more extraordinary mechanical performances compared to those of 2D auxetic fabrics. Ge and Hu [17], for the first time, reported a novel 3D auxetic fabric developed by combining knitting and non-weaving techniques. After fabrication, the fabric structure was converted into a soft composite for stability. Uniquely, this fabric structure was designed to show the NPR effect under compression. Hence, it was recommended for indentation-resistant and impact-resistant applications. Similarly, 3D multi-layer auxetic woven structures having an out-of-plane NPR were developed using the differential modulus of the yarns [30]. The out-of-plane NPR effect is triggered by the Z-direction binding yarns when they are stretched longitudinally. Due to the limitation of the out-of-plane auxetic effect, the authors used their developed fabrics as a reinforcement in polymer composite. Furthermore, the authors claimed improved impact-resistant properties for the 3D auxetic woven fabric reinforced composites. Not long ago, a new milestone was achieved by developing 3D auxetic woven fabrics with in-plane NPR effect [31]. The development is considered to be very significant in view of broader application aspects because it addresses the drawbacks associated with the existing 2D and 3D auxetic textiles. Furthermore, the fabrics that exhibit in-plane auxetic behavior at a higher tensile tension are useful for releasing the contact pressure when they are used in applications such as automotive seat belts, garment belts, etc. Although significant interests have been shown on the design, fabrication, and characteristics of 3D auxetic woven structure, the important information on the fundamental geometrical analytics of structure is yet lacking.

In this study, a geometrical model of 3D woven fabric has been proposed at the micro-level (yarn level) to establish a relationship between the geometrical parameters (in terms of yarn parameters) and the tensile deformation. Since the geometrical analysis is carried out at the micro-level, the geometrical arrangement and the physical properties of yarns are very crucial to facilitate the geometrical model. Thus, a 3D auxetic woven fabric was fabricated to carefully observe the arrangements of the warp and weft yarns, forming auxetic geometry. After observing the geometry in a free state, the fabric was then subject to tensile deformation to examine the behavior of the yarns together with the geometrical deformation of the whole structure. Here, it was found that the structure extends longitudinally and axially under tensile deformation due to its auxetic nature, which causes tension and compression to the weft yarns. Thus, by recognizing the geometry and physical behavior of yarns, semi-empirical equations were drawn based on the geometrical arrangement of the warp and weft yarns, including the tension and compression effect of the yarns. The study shows that the extracted results from the established semi-empirical equations are in good agreement with the experimental results. Therefore, it could be said that the proposed geometrical model will be a useful tool to design, predict, and optimize the auxetic behavior of the 3D auxetic woven fabric.

## 2. Methodology

A comprehensive methodology was adopted to propose an effective geometrical model for the 3D auxetic woven structure. First, the 3D auxetic woven fabric structure was designed and developed with appropriate warp and weft yarns. Then, the fabric was subjected to a tensile loading to observe its deformation behavior at the micro-level and macro-level. Finally, a geometrical model was proposed by establishing semi-empirical equations based on the warp and weft yarn’s geometry at the initial and deformed states.

### 2.1. Structure Design and Fabric Formation

A novel multi-layer 3D auxetic woven structure was previously designed by modifying the conventional 3D orthogonal through-the-thickness structure [31]. The auxetic effect was introduced by featuring an unusual lateral crimp to the warp yarns, which form a special geometry resembling a re-entrant hexagon, as shown in Figure 1a. The unusual lateral crimp of the warp yarns was induced by using two different weft yarn systems, i.e., a coarser, non-elastic yarn and a fine elastic yarn. A solid, coarser yarn was inserted with a 2/2 twill pattern as a binder (through the thickness), creating alternate empty spaces within the warp yarns. Meanwhile, multiple insertions of fine elastic yarn were made in stretched form, which shrank the structure in a relaxed state by forcing the warp yarns to crimp laterally, filling those empty spaces. As a result, an unusual lateral crimp, similar to a sinusoidal wave, was induced in the warp yarns. When this structure is stretched longitudinally, the warp yarns try to become straight, pushing the coarse binding yarns in a lateral direction. Thus, under tensile extension, the width of the structure also increases, and an auxetic effect is achieved. 

For sample fabrication, the three types of polymer-based yarn were procured from Wai Hung Weaving Factory Limited with the desired properties. A single type of yarn made of multi-filament polyester with a 0.7 mm diameter was selected as a warp. Two types of yarn, namely coarse binding yarn and elastic yarn, made of polyester-wrapped polyurethane (PU) having diameters of 2.5 mm and 0.62 mm, respectively, were selected as the weft. In addition, other features of the warp yarn include stability and strength because it will bear tensile loading. Similarly, the coarse binding yarn was solid, but easier to bend to facilitate orthogonal shaping during weaving. A semi-automatic weaving machine with a maximum of 16 heald frames attached to the dobby shedding mechanism was used to produce the fabric sample. The required number of warp yarns were passed through 8 heald frames with a straight drawing-in draft to weave a two-layer 3D woven fabric. A special denting plan, i.e., two yarns per dent followed by an empty dent, was used. This denting sequence kept a gap between the warp yarns which facilitate the insertion of coarse binding yarn. Furthermore, elastic yarn was inserted in a stretch form to shrink the structure along the width direction when the sample was cut off the loom. The developed fabric sample is shown in Figure 1b. 

It is worth mentioning that, in a previous study [31], the 3D auxetic woven structure was investigated with four highly influencing parameters: diameter of binding yarn, bending stiffness of binding yarn, repeats of elastic weft yarn, and stretch percentage of elastic weft yarn. Thereupon, it was concluded that the auxetic behavior of the 3D woven fabrics is primarily associated with the yarn’s properties that limit the weaving process. For example, the higher diameter and stiffness of binding yarn were difficult to weave to achieve a favorable 3D auxetic structure. Therefore, it is preferred to select and develop a fabric sample with the optimized parameters having the maximum auxetic effect for validating the geometric model of the structure because it is considered that a sample with maximum auxetic behavior indicates the best formation of auxetic geometry. The physical properties of the three types of yarn used to develop the sample are given in Table 1. The bending stiffness of the binding yarn was calculated with the beam deflection method reported by Msalilwa et al. [32].

### 2.2. Tensile Test and Measurement of Negative Poisson’s Ratio

To avoid possible slippage during the tensile test, aluminum tabs were mounted on both ends of each sample, covering the gripping area of the clamps. Epoxy EL2 and AT30 hardener were used to mount the aluminum tabs onto the specimen. The narrow-woven fabric samples of width 25 mm and length 150 mm were prepared according to the guidelines provided by tensile testing standard ASTM D5035-11 [33]. Further, the gauge length between the two jaws of the machine was kept at 75 mm. An Instron 5982 universal testing machine (UTM) with a maximum load capacity of 100 kN was used in this experiment. The samples were subjected to a strain-controlled tensile test, where the lower jaw was fixed, and the upper jaw moved at the rate of 30 mm per minute. The test was performed three times for each set of samples, and average values were reported.

The UTM can only provide raw data of the mechanical properties, therefore, a separate digital camera (Canon EOS 800D) was set up in front of the machine to record the change in dimensions of the fabric samples during the tensile deformation, as shown in Figure 2. Thus, each sample was marked with four dots (two longitudinally and two vertically) at a distance of 25 mm before clamping them onto the UTM. The video recording and the tensile test started simultaneously. First, the recorded videos were processed, and pictures were extracted at a rate of 1.5 s, which was equivalent to 1% tensile strain. Then, the extracted pictures were analyzed with screen ruler software to calculate the lateral and longitudinal deformation of the fabric. Finally, the PR for each sample was calculated using Equation (1).
(1)$$\upsilon =-\frac{\varepsilon_{y}}{\varepsilon_{x}}=-\frac{\left(Y^{\prime }-Y_{o}\right)/Y_{o}}{\left(X^{\prime }-X_{o}\right)/X_{o}}=-\frac{\left(Y^{\prime }-Y_{o}\right)X_{o}}{\left(X^{\prime }-X_{o}\right)Y_{o}}$$
where *ε_x_* is strain in longitudinal direction and *ε_y_* is strain in the lateral direction; $Y_{o}$ and $X_{o}$ are the initial width and length as shown in Figure 1b; $Y^{\prime }$ and $X^{\prime }$ are width and length in the stretched state, respectively.

### 2.3. Yarns Analysis

In a previous study [31], it was found that the auxetic effect of the 3D woven fabric is highly associated with the extension of elastic weft yarn and compression of coarse binding yarn during the tensile deformation. Therefore, the inclusion of these two characteristics of yarns are essential for developing a geometrical model.

The extension of elastic weft yarn was simply characterized by performing a tensile test on a single yarn using the Instron 5566 machine. The yarn was fixed between the machine’s jaws, where the lower jaw wax fixed, and the upper jaw moved at a constant velocity of 30 mm/min. Similarly, the compression of coarse binding yarn was determined by performing a compression test. However, conventional flat compression could be applied here because the compression of the binding yarn was applied by the warp yarns. Hence, a customized assembly, as shown in Figure 3, was prepared to perform the test. The assembly was designed in such a way that the two pins, whose diameter was equal to that of the warp yarn, applied the compressive force on the binding yarn. So, the binding yarn was placed between the upper and lower parts of the assembly, perpendicular to the pins. Then, the assembly was placed between the two platens of the Instron 5566 machine. To perform the compression test, the upper platen was moved downward at a constant velocity of 1 mm/min, and the lower platen was fixed. A very sensitive load cell of capacity 50 N was used to obtain precise results.

### 2.4. Geometrical Analysis

The parametric terms used in this geometrical analysis are given below:
$a_{o}$—initial diameter of binding yarn.$b_{o}$—diameter of warp yarn at the initial state and during the first stage of deformation.$b_{ii}$—diameter of warp yarn during the second stage of deformation.$l_{o}$—length of warp yarn between the centers of the two binding yarns at the initial state and during the first stage of deformation.$l_{ii}$—length of warp yarn between the centers of the two binding yarns during the second stage of deformation.$\theta_{o}$—initial inclination angle of warp yarn.$\theta_{i}$—inclination angle of warp yarn in a deformed state.$D_{o}$—sum of the radii of binding yarn and warp yarn at the initial state.$D_{i}$—sum of the radii of binding yarn and warp yarn during the first stage of deformation.$D_{ii}$—sum of the radii of binding yarn and warp yarn during the second stage of deformation.$a_{i}$—height of binding yarn after compression.$a_{i}^{'}$—width of binding yarn after compression.


A geometric model is proposed to predict the NPR effect of the 3D in-plane auxetic woven structure. The following assumptions are made based on the deformation behavior of the fabric during the tensile test:
Figure 4 represent the schematic diagram of the 3D auxetic woven structure while the detailed geometry of the unit cell of the structure is illustrated in Figure 5. It is assumed that all the repeating unit cells are similar in size and shape in the initial state and deformed symmetrically during extension.In the geometrical model, the presentation of an elastic weft yarn has been omitted because the warp and binding yarn analysis is suitable enough to predict the auxetic behavior. However, the effect of the elastic weft yarn, i.e., causing compression to the binding yarn during tensile loading, was included.The initial cross-section of the binding yarn is circular, which, during compression, changes to a race track shape due to the compression of the warp yarns caused by the restoring force of the elastic weft yarns. However, the cross-sectional area of the binding yarn remains constant all the time.The warp yarns are crimped by the binding yarns at the initial state. It is assumed that the part of the warp yarn that is in contact with the binding yarn is circular, while the non-contact part is straight and tangent to the circular part.There is no slippage at the contact points of binding and warp yarns during tensile deformation.During tensile stretching, the structure deforms in two stages: decrimping (Figure 5a) and elongation (Figure 5b) of the warp yarns under tensile loading. Furthermore, it is also assumed that at the decrimping stage, the cross-section of binding yarn changes from a circular one to a race track one, while the diameter of warp yarn remains unchanged. However, in the elongation stage, there is no further change to the dimensions of the race-track-shaped binding yarn, while the diameter of the warp yarn changes due to elongation.
First stage: In the first stage of deformation, the warp yarns start decrimping at a constant rate and become fully straight at a particular tensile strain. In this stage, the length and diameter of the warp yarn remain constant, as shown in Figure 5a. Based on the above assumptions, the geometrical model of the first stage can be re-illustrated in detail, as shown in Figure 6.From Figure 6, the initial distance ($X_{o}$) between the centers of two binding yarns is given by:(2)$$X_{o}=2\bar{pq}+\bar{qs}\;$$The length of $\bar{pq}$ and $\bar{qs}$ in terms of yarn’s parameters can be calculated by solving right angle triangles Δopq and Δqst, respectively.
(3)$$\bar{pq}=D_{o}\sin\theta_{o}$$
(4)$$\bar{qs}=\bar{qt}\cos\theta_{o}$$Whereas:(5)$$\bar{qt}=l_{o}-2\hat{qr}$$
(6)$$\hat{qr}=D_{o}\theta_{o}$$By solving Equations (4)–(6), the following relation can be obtained
(7)$$\bar{qs}=(l_{o}-2D_{o}\theta_{o})\cos\theta_{o}$$Substituting Equations (3) and (7) into Equation (2) gives the following relation
(8)$$X_{o}=2D_{o}\sin\theta_{o}+(l_{o}-2D_{o}\theta_{o})\cos\theta_{o}$$Likewise, the initial height ($Y_{o}$) between the centers of two binding yarns is given as
(9)$$Y_{o}=2\bar{op}-\bar{st}$$To solve Δopq and Δqst, the following relation can be made for $\bar{op}$ and $\bar{st}$
(10)$$\bar{op}=D_{o}\cos\theta_{o}$$
(11)$$\bar{st}=\bar{qt}\sin\theta_{o}$$Substituting Equations (5) and (6) into Equation (11) gives the following equation
(12)$$\bar{st}=\left(l_{o}-2D_{o}\theta_{o}\right)\sin\theta_{o}$$Substituting Equations (10) and (12) into Equation (9) gives the following relation
(13)$$Y_{o}=2D_{o}\cos\theta_{o}-\left(l_{o}-2D_{o}\theta_{o}\right)\sin\theta_{o}$$This relation can be used for calculating the initial height ($Y_{o}$) using the inclination angle ($\theta_{o}$), however, $Y_{o}$ can also be calculated simply by using the following relation.
(14)$$Y_{o}=\frac{a_{o}}{2}+b_{o}$$Similarly, the distance between the centers of the two binding yarns at a deformed state ($X_{i}$), is given below.
(15)$$X_{i}=2\bar{p^{\prime }q^{\prime }}+\bar{q^{\prime }s^{\prime }}+2\bar{n^{\prime }o^{\prime }}$$Whereas:(16)$$\bar{n^{\prime }o^{\prime }}=\left(a_{i}^{'}-a_{i}\right)/2$$By solving the right triangles $\Delta o^{\prime }p^{\prime }q^{\prime }\;$and $\Delta q^{\prime }s^{\prime }t^{\prime }$, the lengths $\bar{p^{\prime }q^{\prime }}$ and $\bar{q^{\prime }s^{\prime }}$ can be expressed with Equations (17) and (18), respectively.
(17)$$\bar{p^{\prime }q^{\prime }}=D_{i}\sin\theta_{i}$$
(18)$$\bar{q^{\prime }s^{\prime }}=\left[l_{o}-2D_{i}\theta_{i}-(a_{i}^{'}-a_{i}\right)]\cos\theta_{i}$$Substituting Equations (16)–(18) into Equation (15) gives the following relation
(19)$$X_{i}=2D_{i}\sin\theta_{i}+[l_{o}-2D_{i}\theta_{i}-(a_{i}^{'}-a_{i})]\cos\;\theta_{i}+(a_{i}^{'}-a_{i})$$The height, $Y_{i},\;$between the coarse binding yarns can express mathematically as,
(20)$$Y_{i}=2\bar{o^{\prime }p^{\prime }}-\bar{s^{\prime }t^{\prime }}$$From $\Delta o^{\prime }p^{\prime }q^{\prime }$ and $\Delta q^{\prime }s^{\prime }t^{\prime }$, $\bar{o^{\prime }p^{\prime }}$and $\bar{s^{\prime }t^{\prime }}$ can be expressed in the following Equations
(21)$$\bar{o^{\prime }p^{\prime }}=D_{i}\cos\theta_{i}$$
(22)$$\bar{s^{\prime }t^{\prime }}=\left(l_{o}-2D_{i}\theta_{i}-(a_{i}^{'}-a_{i}\right))\sin\theta_{i}$$Substituting Equations (21) and (22) into Equation (20) gives the following relation
(23)$$Y_{i}=2D_{i}\cos\theta_{i}-\left[l_{o}-2D_{i}\theta_{i}-(a_{i}^{'}-a_{i}\right)]\sin\theta_{i}$$As both the lateral and longitudinal strains in the initial and deformed states are known, the PR $(\upsilon_{i}$) for the first stage of deformation can be calculated. By putting Equations (8), (13), (19), and (23) into Equation (1), the following relation can be obtained
(24)$$\upsilon_{i}=-\frac{\left[2D_{i}\cos\theta_{i}-\left[l_{o}-2D_{i}\theta_{i}-a_{i}^{\prime }+a_{i}\right]\sin\theta_{i}-2D_{o}\cos\theta_{o}+\left(l_{o}-2D_{o}\theta_{o}\right)\sin\theta_{o}\right]\times \left[2D_{o}\sin\theta_{o}+\left(l_{o}-2D_{o}\theta_{o}\right)\cos\theta_{o}\right]}{\left[2D_{i}\sin\theta_{i}+\left[l_{o}-2D_{i}\theta_{i}-a_{i}^{\prime }+a_{i}\right]\cos\theta_{i}+a_{i}^{\prime }-a_{i}-2D_{o}\sin\theta_{o}-\left(l_{o}-2D_{o}\theta_{o}\right)\cos\theta_{o}\right]\times \left[2D_{o}\cos\theta_{o}-\left(l_{o}-2D_{o}\theta_{o}\right)\sin\theta_{o}\right]}$$Second stage: As the warp yarns become decrimped and become fully straight in the first stage, they then elongate at a constant rate due to the applied tensile loading. In addition, the warp yarn’s diameter will also decrease due to elongation, as shown in Figure 7.



It should be noted that the height and width of the race-track-shaped binding yarns will remain constant, while according to assumption 6, the length and diameter of the warp yarn will change during the second stage of deformation. Therefore, the sum of the radii of warp yarn and binding yarn $D_{i}$ will change to $D_{ii}$, which is given in Equation (25), and the length of warp yarn $l_{o}$ will change to $l_{ii}$, which is written in Equation (26).
(25)$$D_{ii}=\frac{a_{i}}{2}+\frac{b_{ii}}{2}$$
(26)$$l_{ii}=l_{o}+\Delta l$$
where Δl is the change in the length of warp yarn, it can be calculated from the magnitude of tensile extension.

As the warp yarns are decrimped and straightened, the inclination angle ($\theta_{i}$) becomes “0”. By putting $\theta_{i}=0$ into Equations (19) and (23), the following relations can be obtained, respectively
(27)$$X_{ii}=l_{ii}$$
(28)$$Y_{ii}=2D_{ii}$$

As the lateral and longitudinal strains at the second stage of deformation are known, the PR $\upsilon_{ii}$ for the second stage of deformation can be calculated by putting Equations (8), (13), (27), and (28) into Equation (1)
(29)$$\upsilon_{ii}=-\frac{\left[2D_{ii}-2D_{o}\cos\theta_{o}+\left(l_{o}-2D_{o}\theta_{o}\right)\sin\theta_{o}\right]\times \left[2D_{o}\sin\theta_{o}+\left(l_{o}-2D_{o}\theta_{o}\right)\cos\theta_{o}\right]}{\left[l_{ii}-2D_{o}\sin\theta_{o}-\left(l_{o}-2D_{o}\theta_{o}\right)\cos\theta_{o}\right]\times \left[2D_{o}\cos\theta_{o}-\left(l_{o}-2D_{o}\theta_{o}\right)\sin\theta_{o}.\right]}$$


## 3. Results and Discussion

After successfully developing the geometrical model, the first step is to validate it through experimental results. Hence, the model can be used to predict, present, and analyze critical findings based on different structural parameters of the 3D auxetic woven fabric.

### 3.1. Experimental Observation

#### 3.1.1. Determining the Poisson’s Ratio during the First Stage of Deformation

To validate the geometrical model, the calculated tensile strain and PR results should be based on the same geometrical parameters as that of the developed 3D woven fabric sample. Considering that, all the parameters in the initial state are reasonably straightforward to obtain. For example, the radii of the coarse binding yarn ($\frac{a_{o}}{2}$) and warp yarn ($\frac{b_{o}}{2}$) can be taken from Table 1, the initial distance ($X_{o}$) between two coarse binding yarns can be determined by dividing the total length of a sample by the total repeats of coarse binding yarn, and the initial height ($Y_{o}$) can be calculated using Equation (14). When these parameters are known, the initial inclination angle ($\theta_{o}$) and length ($l_{o}$) of the warp yarn can be calculated using Equations (8) and (13).

In contrast, other parameters, particularly the compression effect of coarse binding yarn in a deformed state, are comparatively complicated to calculate. According to assumption (3), the cross-section of binding yarn changes to a race track shape during tensile deformation due to the compression of warp yarns caused by the restoring force of the elastic weft yarns. Therefore, the magnitude of force applied by elastic weft yarn should be known. During tensile deformation, the force applied by elastic weft yarn increases because of the lateral expansion of the structure. So, the first step is to calculate the amount of lateral deformation of the structure without assuming the coarse binding yarn’s compression. For this purpose, Equations (8), (13), (19), and (23) can be used, while considering $a_{i}^{'}=a_{i}=a_{o}$. When the percentage of lateral deformation of the structure is known, the magnitude of force applied by elastic weft yarn can be calculated from the force–strain curve of elastic weft yarn as shown in Figure 8. Eventually, the compression caused to the coarse binding yarn at a given force can be determined from the force–compressive extension curve of the coarse binding yarn, as shown in Figure 9.

After calculating the compression of coarse binding yarn in a deformed state, other parameters such as the final height ($a_{i})\;$and width ($a_{i}^{'}$) of race-track-shaped binding yarn can be calculated based on assumption 3. Now, using Equation (19), the inclination angle ($\theta_{i}$) can be calculated at a certain tensile deformation ($X_{i}$). Thus, all the variables are known, therefore, the PR ($\upsilon_{i}$) during the first stage of deformation can be determined using Equation (24). The calculated values of PR for the first stage of deformation are given in Table 2.

#### 3.1.2. Determining the Poisson’s Ratio during the Second Stage of Deformation

The inclination angle ($\theta_{i}$) reaches 0 at a tensile strain of 8.07%, which indicates that the warp yarn is decrimped and becomes fully straight. At this point, the second stage of deformation starts, where no changes occur to the other parameters except to the length and diameter of the warp yarn. This specific point is determined as “critical strain,” and the PR is called “critical Poisson’s ratio”. After the critical strain, the warp yarns elongate at a rate of tensile strain, which is a known parameter. However, the decrease in the diameter of the warp yarn ($b_{ii}$) at a particular tensile strain needs to be calculated. So, a customized tensile test was performed on the warp yarn, in which the cross head of UTM machine needed to be stopped after every 1% strain for 10 s to observe the diameter of the warp yarn using a thickness gauge. The variation trend of warp yarn’s diameter ($b_{ii}$) as a function of tensile strain (ε) is shown in Figure 10. It can be observed that the change in $b_{ii}$ has a polynomial trend, therefore, according to a second-degree polynomial trend, the following equation can be established
(30)$$b_{ii}=m_{1}\varepsilon_{i}^{2}+m_{2}\varepsilon_{i}+m_{3}$$
where $\varepsilon_{i}$ is the tensile strain, and $m_{1}$, $m_{2}$, and $m_{3}$ are the constants that can be determined from the experimental resutls of the warp yarn. Substituting Equation (30) in Equation (25) gives the following equation
(31)$$D_{ii}=\frac{a_{i}}{2}+\frac{m_{1}\varepsilon^{2}+m_{2}\varepsilon +m_{3}}{2}$$

From Equation (31), $a_{i}$ is a known parameter, whereas the three other unknown constants were calculated experimentally from the relationship of warp yarn’s diameter and the tensile strain, which are: $m_{1}=0.003$, $m_{2}=-0.0102$, and $m_{3}=0.6608$.

When the sum of radii of the binding yarn and warp yarn $D_{ii}$ are known at a particular tensile extension, $X_{ii}$, then the PR at the second stage of deformation can be calculated using Equation (29). Finally, the yarn’s parameters, the structural variables of the geometric model, and the calculated PR are given in Table 2.

It should be noted that if the tensile strain is less than the critical tensile strain, then $\upsilon_{i}$ (Equation (24)) will be used to calculate the PR, otherwise $\upsilon_{ii}$ Equation (29) will be used.

Figure 11 presents the PR curves plotted against the tensile strain of the developed 3D auxetic woven fabric and the calculated results obtained from the geometrical model. It can be seen that both the experimental and calculated PRs follow a similar trend. The agreement of calculated and experimental results confirms the reliability of the geometrical model. The PR first decreases, reaches a minimum at 8.07% tensile strain, and then follows a steadily increasing trend. From the analysis of the structure’s geometry, it is clear that the lateral crimp of warp yarn plays an important role in the auxetic behavior under the tensile extension. The warp yarn starts decrimp under the tensile deformation, and the PR decreases. The effect of decrimping continues until the warp yarn becomes fully straight, which takes effect at 8.07% tensile strain. Hereafter, there is no decrimping of warp yarn that indicates there will be no lateral extension with further tensile extension. In addition, the diameter of warp yarns decreases due to longitudinal extension. As a result, the PR increased beyond 8.07% tensile strain. 

### 3.2. Prediction of Auxetic Behavior

#### 3.2.1. Effect of Binding Yarn Compression

A previous study [31] claimed that the coarse binding yarn becomes compressed during tensile extension. From the experimental assessment of the structure, it is quite clear that compression occurs due to the restoring force of elastic yarn. However, what could not be answered is how much the compression of binding yarn effects the auxetic behavior. In comparison, the geometrical model can calculate the effect of binding yarn compression on the auxetic behavior of 3D woven fabric. Figure 12 shows the PR versus tensile strain curves. The two curves, with compression (W-C) and without compression (W/O-C), are calculated using the geometrical model by keeping all the parameters constant, except for the compression effect of the binding yarn. For WO-C, it is supposed that the diameter of the binding yarn ($a_{o}$) does not change during the tensile strain, while for W-C, the diameter of the binding yarn decreases as the structure elongates in both the longitudinal and lateral directions. It is worth mentioning that the compression that occurred to the binding yarn is calculated experimentally, as explained in Section 3.1. From Figure 12, it can be observed that the auxetic behavior of W-C is 54% less than that of WO-C. This means that the binding yarn’s compressive stiffness is critical and must be considered actively when one is developing 3D auxetic woven fabric.

#### 3.2.2. Effect of Binding Yarn Diameter

It is well known that [17] the diameters of two yarns (warp and binding yarns) significantly affect the NPR of 3D auxetic structures. Later on, it was proven again through an experimental study [31]. Since then, the effect of diameter of yarns on auxetic behavior has been understood, however, a scientific reason for the difference is still lacking. To provide a concrete reason, PR of the 3D woven structure was calculated using the geometrical model. Three different binding yarns diameter, i.e., 1.9 mm, 2.28 mm, and 2.75 mm, were used, while other parameters were kept constant. The PR–tensile strain curves of the 3D woven fabrics based on the different diameters of binding yarns are shown in Figure 13.

It can be found that the auxetic effect increases by increasing the diameter of the binding yarn, which validates the literature. For a solid reason, the numerical values generated by the geometrical model were analyzed. It is concluded that the initial lateral crimp percentage (%) of warp yarn decreases by increasing the diameter of the binding yarn. The crimp% can be calculated using Equation (32). For binding yarn diameters 1.9 mm, 2.28 mm, and 2.75 mm, the crimp% are 8.086, 8.011, and 7.949, respectively. Less lateral crimp in the warp yarn means the lateral expansion of auxetic structure will be triggered at a lower longitudinal strain, and the structure will reach its maximum lateral deformation under less strain. Since the longitudinal strain is inversely proportional to the NPR (Equation (1)), the auxetic behavior of the structure with the warp yarns being less laterally crimped is less notable, and vice versa.
(32)$$Crimp\;\%=\frac{l_{o}-X_{o}}{X_{o}}\times 100$$

#### 3.2.3. Effect of Binding Yarn Spacing

Similar to the effect of binding yarn compression and binding yarn diameter, the spacing between the two binding yarns is also important to assess the auxetic behavior of the 3D auxetic woven structure. For this assessment, a similar methodological approach was applied, i.e., all the other parameters were kept constant, and the initial distance between the two binding yarns ($X_{o}$), also referred to as binding yarn spacing, was changed for three measurements. The three selected values for $X_{o}$ were 2.8 mm, 3 mm, and 3.2 mm. The PR values were then calculated using the developed geometrical model that are presented in Figure 14.

It can be explained that the binding yarn spacing has a direct relationship with the NPR effect, i.e., the NPR increases by increasing the yarn spacing ($X_{o})$, and vice versa. After evaluating the measured values of each parameter, it was found that the binding yarn spacing also effected the lateral crimping of the warp yarn. Higher yarn spacing caused less crimping of the warp yarn, producing a high NPR effect at a lower longitudinal strain value. The calculated lateral crimp values are 9.74 mm, 8.01 mm, and 6.97 mm when $X_{o}$ are 2.8 mm, 3 mm, and 3.2 mm , respectively.

## 4. Conclusions

The 3D woven fabric was designed and fabricated with in-plane auxetic behavior. The auxetic effect was visualized by the unusual lateral crimping of warp yarns that form a special geometry, resembling a re-entrant hexagon. Based on the geometrical arrangement of yarns in the structure, a micro-level geometrical model was proposed in terms of the yarn parameters to establish a relationship between PR and tensile strain through semi-empirical equations. The following conclusions can be drawn according to the study.
As the geometrical model is based on the micro-level analysis of the structure, therefore, it is useful to predict the auxetic behavior with given yarn parameters. Additionally, it can help to find specific reasons for the NPR effect in relation to a particular parameter.Good agreements are observed between the experimental results and the results obtained from the established empirical equations based on the geometrical analysis.It was found from the geometric analysis that the compression that is applied to binding yarns during lateral expansion significantly affects the NPR of the structure, i.e., the NPR decreases by 62.5% after the compression of the coarse binding yarn. Therefore, the compressive stiffness of binding yarn should be considered carefully.The diameter of binding yarn and the spacing between the two binding yarns affect the lateral crimp percentage of the warp yarn. Both the diameter and spacing of binding yarns have an inverse relationship with the crimp percentage of the warp yarn. For example, the PR of the samples are −2.1 and −3.0 when the binding yarn spacing is 2.8 mm and 3.2 mm, respectively.The lateral crimp of warp yarn has a strong relationship with the NPR of the structure. The NPR increases if the crimping degree is lower and decreases if the crimping degree is higher. For example, when the lateral crimping of warp yarn is 8.09%, the PR is −2.0, while the PR for a lateral crimping of 7.95% is −3.78.

The retainability of the auxetic effect is of great significance to its practical application. Therefore, the auxetic behavior of the fabrics under repetitive tensile forces can be evaluated experimentally. Furthermore, the current study uses coarse yarns to produce the fabric. The auxetic fabric can be used for broader applications if the fabric is developed with finer yarns.

## Figures and Tables

**Figure 1 polymers-15-01326-f001:**
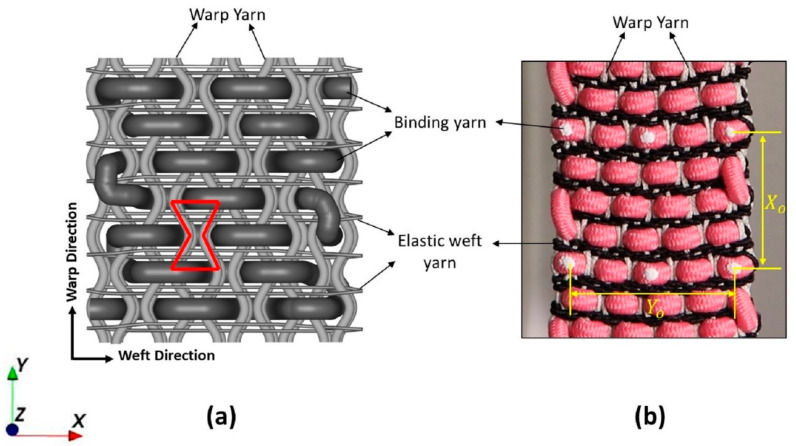
Three dimensional auxetic woven structure: (**a**) schematic representation; (**b**) real fabric.

**Figure 2 polymers-15-01326-f002:**
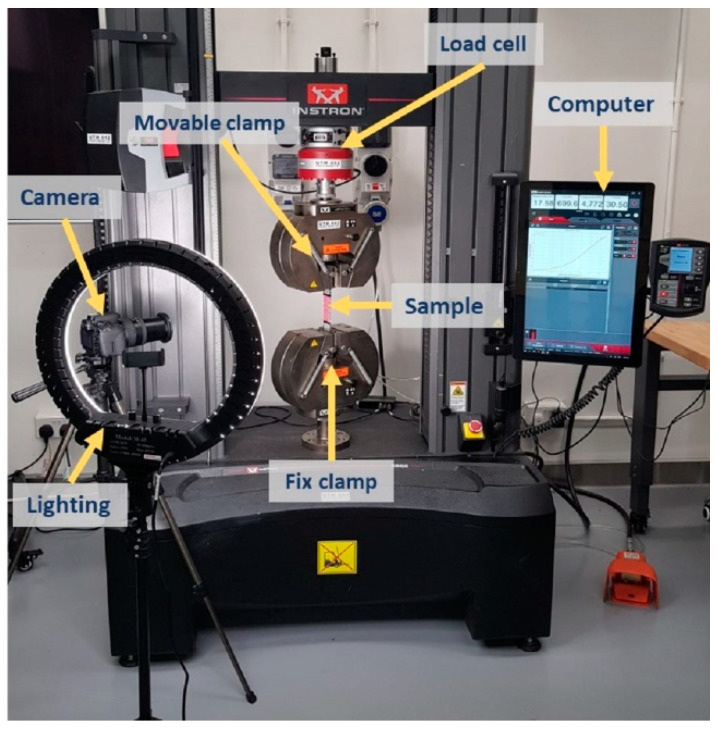
Testing setup on a universal testing machine for Poisson’s ratio measurement.

**Figure 3 polymers-15-01326-f003:**
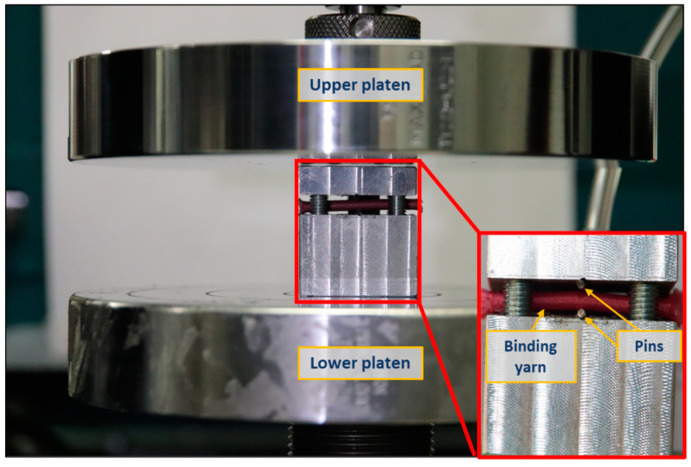
Binding yarn compression test assembly.

**Figure 4 polymers-15-01326-f004:**
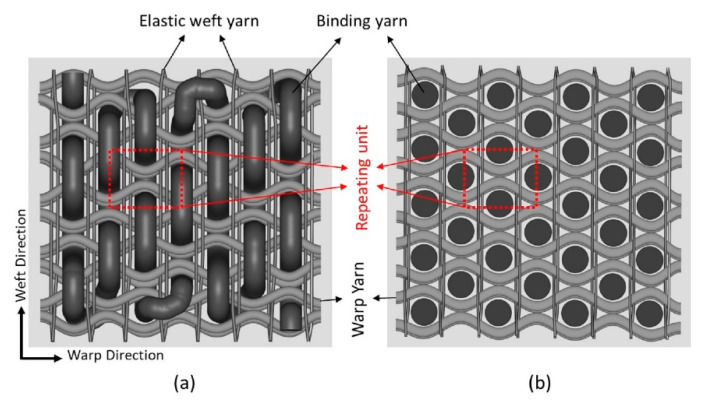
Schematic diagram of 3D auxetic woven structure: (**a**) top view; (**b**) top cross-sectional view [31].

**Figure 5 polymers-15-01326-f005:**
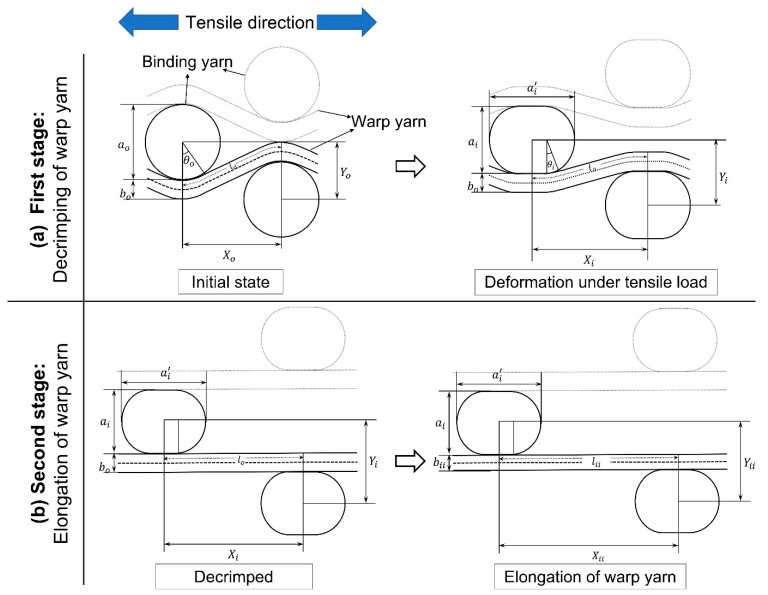
Geometrical model of 3D auxetic structure: (**a**) decrimping of warp yarns under tensile load; (**b**) elongation of warp yarns under tensile load.

**Figure 6 polymers-15-01326-f006:**
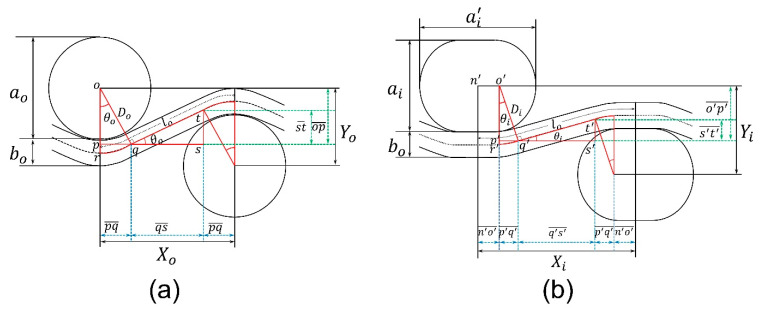
First stage of deformation of a unit-cell: (**a**) initial state; (**b**) deformed state.

**Figure 7 polymers-15-01326-f007:**
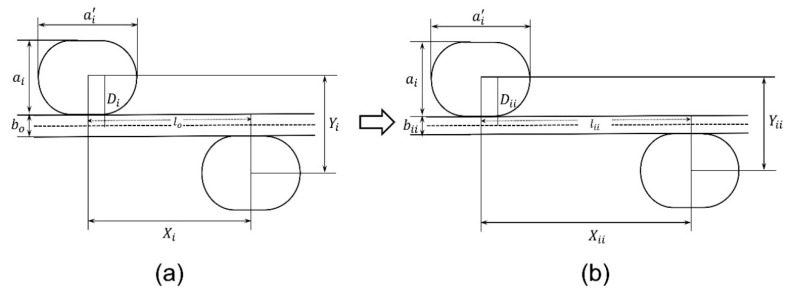
Second stage of deformation of a unit cell: (**a**) decrimped and (**b**) elongation of warp yarn.

**Figure 8 polymers-15-01326-f008:**
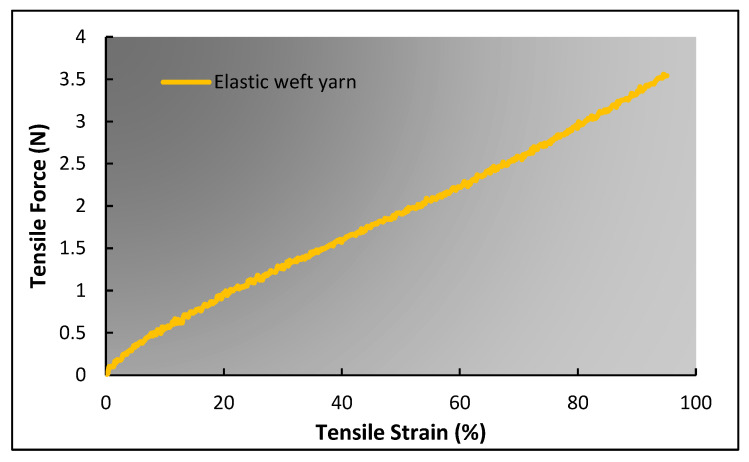
Tensile force–strain curve of elastic weft yarn.

**Figure 9 polymers-15-01326-f009:**
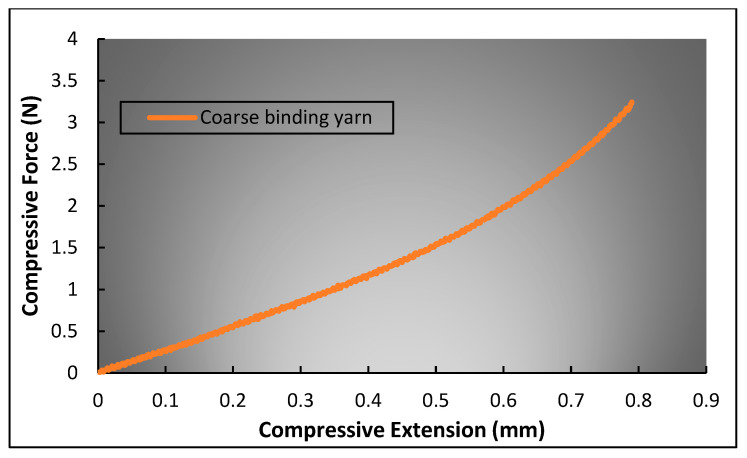
Compressive force–extension curve of coarse binding yarn.

**Figure 10 polymers-15-01326-f010:**
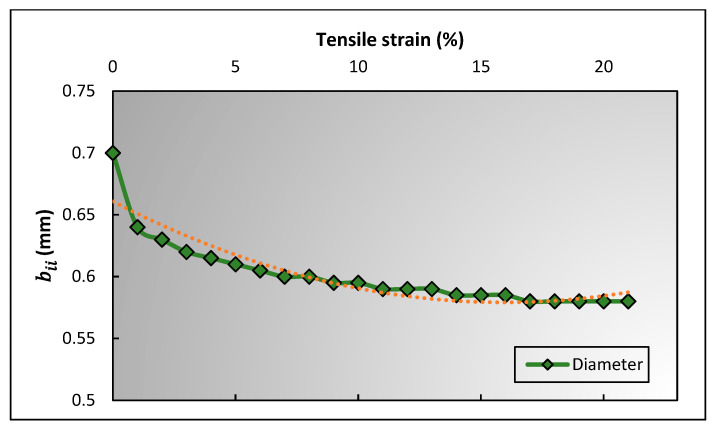
Trend of a diameter of warp yarn against tensile strain.

**Figure 11 polymers-15-01326-f011:**
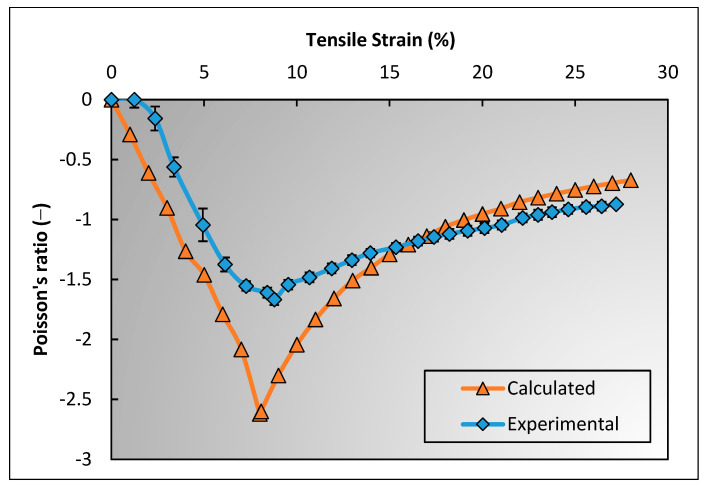
Comparison of calculated and experimental results of Poisson’s ratio against tensile strain.

**Figure 12 polymers-15-01326-f012:**
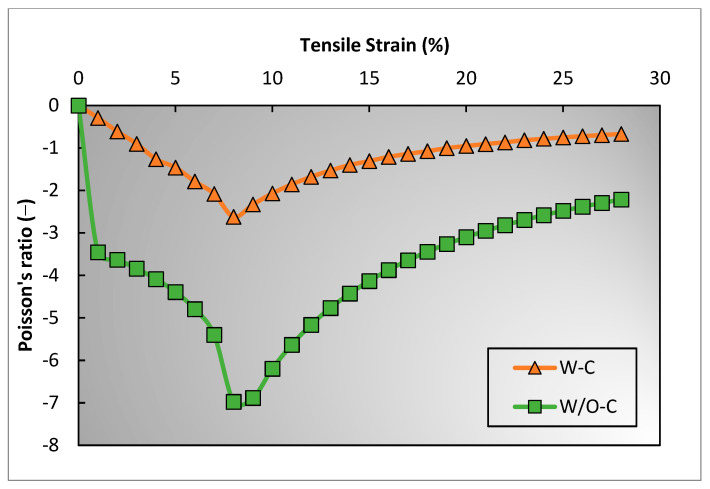
Effect of binding yarn’s compression on auxetic behavior of 3D woven structure.

**Figure 13 polymers-15-01326-f013:**
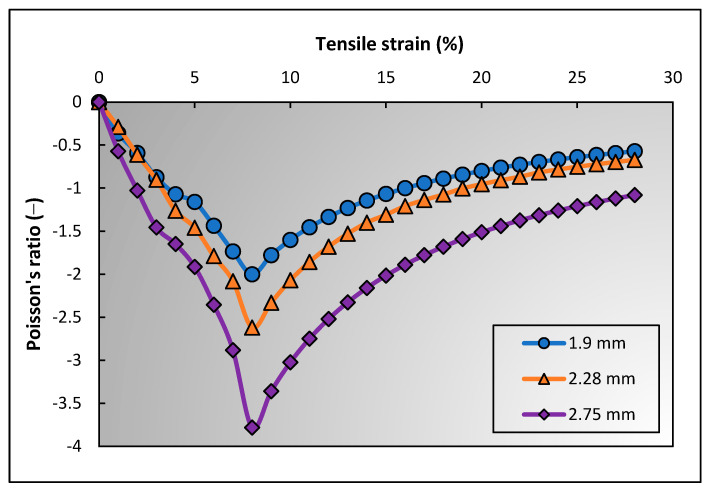
Effect of binding yarn diameter on auxetic behavior of 3D woven structure.

**Figure 14 polymers-15-01326-f014:**
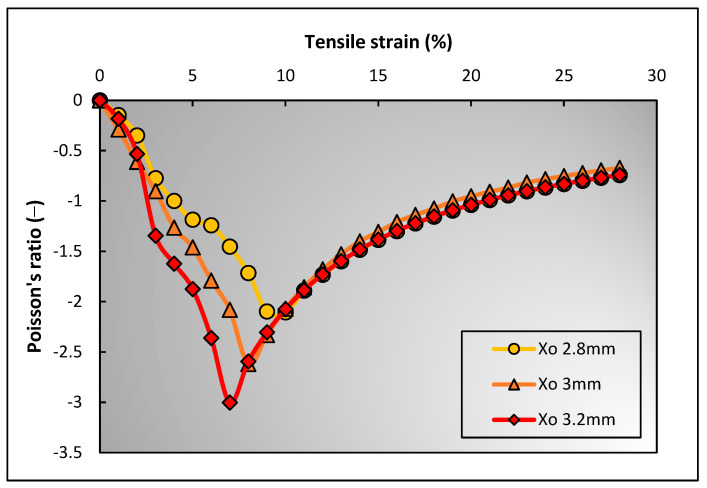
Effect of binding yarn spacing on auxetic behavior of 3D woven structure.

**Table 1 polymers-15-01326-t001:** Properties of the yarns used for sample preparation.

Yarn	Material	Type	Diameter (mm)	Tensile Modulus (MPa)	Tensile Strength (N)	Bending Stiffness (×10^−6^ Nm^2^)
Warp yarn	Polyester multi-filament	Braided yarn	0.70	153	106	-
Binding yarn	Polyester-wrapped PU	Braided yarn	2.28	8	203	0.66
Elastic weft yarn	Polyester-wrapped PU	Braided yarn	0.62	3	14	-

**Table 2 polymers-15-01326-t002:** Structural parameters and the calculated Poisson’s ratio results of 3D auxetic woven fabric.

State		Tensile Strain (%)	$a_{o}$$\;\mathrm{or}\;a_{i}$ (mm)	$b_{o}$$\;\mathrm{or}\;b_{ii}$ (mm)	$D_{o}$$\;\mathrm{or}\;D_{i}$$\;\mathrm{or}\;D_{ii}$ (mm)	$X_{o}$$\;\mathrm{or}\;X_{i}$$\;\mathrm{or}\;X_{ii}$ (mm)	$Y_{o}$ or Y (mm)	$\theta_{o}$$\;\mathrm{or}\;\theta_{i}$ (rad)	$l_{o}$$\;\mathrm{or}\;l_{ii}$ (mm)	PR
**Initial state**		0	2.28	0.70	1.490	3.000	1.840	0.460	3.242	0
**Under tensile deformation**	First stage	1	2.19	0.70	1.447	3.030	1.845	0.436	3.242	−0.29
		2	2.12	0.70	1.410	3.060	1.862	0.408	3.242	−0.61
		3	2.05	0.70	1.377	3.090	1.890	0.376	3.242	−0.90
		4	2.00	0.70	1.351	3.120	1.933	0.337	3.242	−1.26
		5	1.93	0.70	1.316	3.150	1.974	0.295	3.242	−1.46
		6	1.87	0.70	1.286	3.180	2.037	0.243	3.242	−1.79
		7	1.78	0.70	1.245	3.210	2.108	0.177	3.242	−2.08
		8	1.62	0.70	1.158	3.240	2.226	0.047	3.242	−2.61
		**8.07**	**1.62**	**0.70**	**1.113**	**3.242**	**2.226**	**0**	**3.242**	**−2.59**
	Second stage	9	1.62	0.62	1.110	3.270	2.221	0	3.27	−2.33
		10	1.62	0.61	1.108	3.300	2.216	0	3.3	−2.07
		11	1.62	0.60	1.105	3.330	2.211	0	3.33	−1.86
		And so on								

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
