# Peer review of "Geometric Analysis of Three-Dimensional Woven Fabric with in-Plane Auxetic Behavior"

_polymers, 2023, doi:10.3390/polym15051326_

Round 1
Reviewer 1 Report
This paper contains extensive studies and data on geometrical analysis based on semi-empirical equations of 3D auxetic woven fabric structures. In this study, the auxetic effect of the 3D woven fabric was developed with a special geometry. The paper is very well written and has values insight that can be useful for the scientific community and industry. The methodology is scientific, and the article is organised and written according to the standards of the journal, with the exception of some minor grammatical errors in the sentence structure. The article lacks a sufficient review of recent literature. I suggest a minor revision before considering publication.
Reviewer 2 Report
This study reports “Geometric analysis of three-dimensional woven fabric with in-plane auxetic behavior”. The authors proposed a geometrical model to validate and predict the negative Poisson’s ratio of 3D woven fabrics. I found theresearch work interesting; it reports a mathematical model to validate the experimental work. Before considering it for publication, the authors should address the following comments to further improve the paper:
1. The Abstract is too short. The background of the study should be included at the start of the Abstract. The keywords are long, keep them more specific to the study’s domain.
2. Avoid writing short paragraphs, specially in the Introduction part. For example, the first two paragraphs can be merged as one.
3. Line-133 & 311, in scientific writing, the use of First-Person Pronouns is not considered good. It should be written in passive voice.
4. The authors mentioned multiple times “in previous study” however they did not give a reference to it. For example, lines168, 208, etc.
5. In Table 1, the yarn’s bending stiffness value of “0.66” is taken from reference [31]? Please clarify.
6. Figure 4 seems to be from the previous study, if so, it should be properly cited. Consider it for all Figures/tables.
7. Page 10, Equation (24) is incomplete. Some Equations have improper brackets, please check all carefully.
8. Results and Discussion cannot be sub-heading of methodology. Make corrections to other headings as well.
9. The font size, font style, and formatting of Figure 8 and Figure 9 are different from other Figures. The authors should show uniformity in formatting the Figures, Tables, and language.
10. Section “authors contribution” is not correct. Sample development and methodology fall in the same category. Further, the conceptualization of the study is not mentioned for any author.
Reviewer 3 Report
The authors studied the geometric analysis of three-dimensional woven fabric with in-plane auxetic behavior. The manuscript is well-written, and the paper has a scientific contribution to the field of auxetic textiles, which is an emerging field of the future. However, the article has serious shortcomings which need to be addressed before further processing in this journal.
1 – In line 41, the authors mentioned “there are only two techniques for introducing auxetic property into textile 41 fabrics.” However, the techniques are not limited to only two. So, it's better to remove “only”.
2 – In line 90, do not start a paragraph/sentence with “also”.
3 – In line 185, the ASTM standard has not been cited.
4 – In line189-191, The sentence “the test was repeated three times for each set of samples to take an average and report the reproducibility of the results.” is not clear.
5 – The authors calculated the Poisson’s ratio by recording a video of tensile test. Why did not they use an extensometer? Its easier and more accurate.
6 – Figure 3, is inappropriate, it will be better if the zoom portion is adjusted on the top-right or bottom-right of the main image.
7 – Figure 5, “Initial State” the “S” should be small. Please make corrections.
8 – Figure 5 seems blur. Check the Figures quality, the DPI should be according to the journal guidelines.
9 – Figure 6 and Figure 7 have two sub schemes; therefore, it should be numbered as Figure x (a) and Figure x (b), respectively.
10 – Equation-24 is incomplete, if the reason is font size, then reduce the size to the minimum according to the journal guidelines.
11 – The Results and discussions heading is wrong, it should be under heading 1.
12 – Figure 12, the unit of tensile strain is missing. If the Poisson’s ratio has no unit, the put an empty bracket with hyphen for example (-).
13 – In line 439, replace “before all else” with “It is well known that”.
14 – Line 440, replace “Following” with “Later on”.
15 – The conclusion part should be revised. The authors should provide numbers to support the statements. For example, the authors claimed that “The diameter of binding yarn and the spacing between the two binding yarns affect the lateral crimp percentage of the warp yarn.” How much is the affect in terms of percentage?
16 – In line 492, replace “the geometrical analysis suggests” with “it was found from the geometric analysis”.
17 – In line 500, do not write the word “vice versa” write a complete sentence.
18 – The authors should discuss future recommendations for the conducted research study.
Reviewer 4 Report
why you repeat every figure twice ? and put red line in one of them.
in figure 10 the red line indicate to what?
Reviewer 5 Report
Manuscript Number: polymers-2199585
Title: Geometric analysis of three-dimensional woven fabric with in-plane auxetic behavior
Article Type: Article
In the manuscript the experimental research as well as the mathematical modelling of the three-dimensional woven fabric with in-plane auxetic behavior is presented. The topic is intensively researched nowadays. According to the Web of Science the “auxetuc” phrase was used only in 78 publications in 2012 whereas in 2022 one could find 524 papers. Authors published a paper with a very similar title:
AU Zeeshan, M; Hu, H; Zulifqar, A
TI Three-dimensional narrow woven fabric with in-plane auxetic behavior
SO TEXTILE RESEARCH JOURNAL
SN 0040-5175
DI 10.1177/00405175221109639
EA JUN 2022
Unfortunately, I do not have access to this journal and for that reason I cannot tell what is the connection between these two publications.
The manuscript is well written. As I can see it is already after the first round of reviews. Therefore my only recommendation is to add the standard deviation bars to all graphs in which the experimental results are presented.
Round 2
Reviewer 2 Report
The comments are well addressed. The paper can be accepted now.
Reviewer 3 Report
The authors incorporated all my comments. However, I still have one more minor suggestion for improvement as follow:
1. Please remove the heading of Future recommendation. The below text without heading 5 is fine.
